# Effectiveness of Smart Applications of Oral Hygiene Instructions on Periodontal Health Compared to Oral Information: A Prospective Study

**DOI:** 10.3390/ijerph192214703

**Published:** 2022-11-09

**Authors:** Dhafer Alasmari, Mazen Sulaiman Alkhalaf, Saeed Maeed Alqahtani, Nasser Raqe Alqhtani, Abdullah Saad Alqahtani, Khalid Gufran, Yasser Khaled Alotaibi

**Affiliations:** 1Department of Periodontology and Oral Medicine, College of Dentistry, Qassim University, Buraydah 52571, Saudi Arabia; 2College of Dentistry, Qassim University, Buraydah 52571, Saudi Arabia; 3Consultant Periodontics, Dental Department Border Guard Center, Riyadh 12211, Saudi Arabia; 4Department of Oral and Maxillofacial Surgery and Diagnostic Sciences, College of Dentistry, Prince Sattam Bin Abdulaziz University, Al-Kharj 11942, Saudi Arabia; 5Department of Preventive Dental Sciences, College of Dentistry, Prince Sattam Bin Abdulaziz University, Al-Kharj 11942, Saudi Arabia; 6Consultant Periodontist, Prince Sultan Military Medical City, Riyadh 12233, Saudi Arabia

**Keywords:** oral health instruction, smartphone application, periodontal health, plaque index, gingival index, patients’ compliance

## Abstract

Nowadays, smartphone applications are widely used in different areas of life, including medical science. The present study aimed to assess the effectiveness of a smartphone application of oral hygiene instructions (OHI) on periodontal health compared to the traditional chairside OHI. All the participants (*n* = 39) were divided into two groups: control group and test group. Participants of the control group were given verbal OHI and participants of the test group were asked to use a smartphone application to upkeep their oral habits. The gingival index (GI) and Quigley–Hein Turesky modification index (QHTMI) were used for scoring the plaque accumulation at baseline and after one month of the periodontal treatment. A paired *t*-test and an independent *t*-test were performed to compare the mean of GI and QHTMI between intra-group and inter-group, respectively. The paired *t*-test showed significant differences in GI and QHTMI improvement after one month in both groups. Moreover, the independent *t*-test showed no significant differences between the two groups. However, the test group showed a lower mean score in GI and QHTMI assessments compared to the control group. Smartphone applications in oral health applications did not exhibit any superiority in periodontal health over the traditional OHI method in the current study.

## 1. Introduction

During any dental treatment, one of the most significant factors is maintaining good oral hygiene. Attaining ideal oral health through preventive measures is the primary concern of dental health practitioners. Dentists and dental hygienists encourage patients to practice proper oral care by themselves; therefore, patients are advised to follow a benchmark or regimen of oral care [1]. In any case, the success of periodontal treatment is dependent on patients’ aptitude to uphold good oral hygiene. Many previous studies reported an increase in caries incidence in the root areas due to the lack of patient compliance after the periodontal treatment [2,3,4].

Any periodontal diseases and dental caries are caused by an organized form of biofilm called dental plaque [5]. Inflammation is caused when the dental plaque accumulates around and under the gingiva, which eventually destroys the gingival tissue and leads to complicated periodontal conditions if the accumulation of dental plaque is untreated over days or weeks [6,7]. Gingivitis is more prevalent during puberty and poses a greater risk of evolving intricate gingival conditions [8]. Therefore, controlling dental caries and gingivitis is one of the major challenges for dentists.

Motivating patients to maintain good oral hygiene is important to achieve optimum oral health care. Brushing the teeth and flossing are the main mechanical methods that should be performed by patients. Dentists need to motivate and encourage patients to implement these habits properly on a regular basis [9,10]. However, patients oftentimes fail to maintain the oral hygiene regimen which was instructed to them during the chairside time due to their hectic lifestyles in the current fast-paced time.

Smartphones are extensively used in every aspect of life in the current era, including in health sciences. Medical practices from teaching, research, and patient care to diagnosis of diseases can be performed easily nowadays using a handheld smartphone via different applications. In recent years, smartphone applications are often used in dentistry as an adjunct apparatus for motivation and oral health education [11,12,13,14,15]. Using different applications and text messaging services makes it simpler to motivate patients to maintain their oral care. In addition, along with smartphone applications, different social media platforms are also motivating patients to improve their oral health [16]. The applications are carefully developed and designed for use in leisure time and provide advantages to patients who can use their sense of control to upkeep better oral health [17,18]. Different smartphone applications are available that send notifications to patients to remind them to take care of their teeth [19,20]. Moreover, different messaging apps are also used widely to remind patients to maintain their oral health care [21,22]. Hence, the current study aimed to assess the effectiveness of smartphone applications compared to the old traditional methods in terms of oral hygiene instruction.

## 2. Materials and Methods

This prospective study was conducted in the College of Dentistry, Qassim University, Saudi Arabia. Participants for this study were selected from patients who visited the outpatient department (OPD) in the College of Dentistry at Qassim University. A total of 39 patients were selected by the convenience sampling technique and gave consent to participate in this study. The study was conducted in the male section of the university; therefore, all the participants partaking in this current study were male.

After performing the periodontal diagnosis, scaling, and root planing, the patients were divided randomly into two groups (control group and test group). The control group consisting of 19 participants was given verbal oral hygiene instructions, and the test group consisting of 20 participants was asked to download a free smartphone application for their oral hygiene instruction. The application (Healthy Teeth–Tooth Brushing Reminder with timer) was used as a reminder for tooth brushing twice a day. They received a notification from the application to remind them to brush. The application has a separate menu with a graphics interchange format (GIF) to demonstrate how to brush properly. Moreover, the application also sent notifications on when to change their brush, which coincided with their next visit to the dentist.

The gingival index (GI) [23] and Quigley–Hein Turesky modification index (QHTMI) [24] for scoring plaque accumulation (Table 1) were recorded in all the patients in two different stages: at baseline (T0) and after one month of the treatment (T1). A disclosing solution was used to determine the plaque accumulation. One single examiner assessed all the patients to avoid any conflict. Eight randomly selected patients (20% of the total sample size) were assessed twice at T0 and T1 stages for the intra-examiner reliability examination.

Inclusion criteria for this study were: patients aged from 18 to 60 years, good general health, a minimum of 20 permanent teeth except for third molars, and patients who required scaling and root planing regardless of the GI and QHTMI scores. Moreover, patients had to be diagnosed with generalized periodontitis, corresponding to stage I and stage II of the world classifications of 2018 [25]. On the other hand, patients with orthodontic treatment, presence of removable partial dentures, advanced periodontal diseases, use of antibiotics three months prior to entry into the study, non-surgical treatment (scaling) during the past three months before baseline examination, and a history of allergies to oral care products were excluded from the current study. In addition, the smartphone application used in the current study could only be used on Android and iOS phones; therefore, participants who were using other operating systems were also excluded from this study.

### Statistical Analysis

Statistical analyses were performed using the statistical software IBM SPSS, version 27 (IBM Co., Armonk, NY, USA). Intra-class correlation (ICC) was used to identify the intra-class reliability. ICC agreement is considered excellent if the value of ICC is 0.75–1, good if ICC is 0.60–0.74, fair if ICC is 0.40–0.59, and poor if ICC is less than 0.40 [26]. The Kolmogorov–Smirnov test was performed to assess the normal distribution of the data. A paired *t*-test was conducted to observe the mean difference between GI and QHTMI within the group and an independent *t*-test was performed to perceive the mean difference between GI and QHTMI between the two groups. A *p*-value < 0.05 was considered statistically significant.

## 3. Results

ICC statistics showed an excellent correlation between the intra-examiner reliability for the measurement of the GI and QHTMI. The Kolmogorov–Smirnov test showed that data were normally distributed. Therefore, parametric tests were used for this study. A total of 39 participants (control group = 19 and test group = 20) with a mean age of 40 years were included in this study. For comparing the homogeneity of the participants recruited in each group, an independent *t*-test was performed for the mean difference of the GI and QHTMI at stage T0 between the two groups, which showed no significant difference (*p* > 0.05) (Table 2). In addition, the independent *t*-test did not identify any significant differences (*p* > 0.05) in the GI and QHTMI scoring system after one month (T1) in the traditional oral health instruction group and smartphone instruction group. Yet, the smartphone instruction group showed a lower mean score of the GI and QHTMI than the traditional instruction group (Table 2). For comparing the before and after periodontal health status, a paired *t*-test showed significant differences (*p* < 0.0001) in the GI and QHTMI scoring systems between the T0 and T1 stages in both groups (Table 3).

## 4. Discussion

Oral health plays an important role in the overall health status of each individual. Therefore, taking proper care of oral health is mandatory. Though dentists provide the optimal treatment to the patients and enormous instruction during the chairside time on how to practice good oral care, it depends on patients’ cooperation. With the advancement of technology, healthcare facilities are becoming ever more accessible. However, there is still a chance to improve these facilities in the field of dentistry [21].

The dental literature has reported previously on the benefits of utilizing technologies as a tool in different aspects of dental sciences [27,28,29]. As young people have widely accepted smartphones in their daily lives, it could be a recognized tool for dental health education [29,30,31]. Moreover, different dental education applications were reported to be more interactive by patients [32]. Educational applications could deliver a mass of health information to a wider audience because of the popularity and advancement of smartphones in the current era [14,26,30,33]. These applications might not increase dental knowledge; however, users could be motivated to practice better oral care regimens [18]. In this study, the effectiveness of smartphone applications in terms of oral hygiene instruction compared to the traditional chairside instruction method was assessed in periodontal treatment. The current study exhibited no substantial differences in the periodontal health status between those who used smartphone applications and those who followed the chairside instructions in their daily oral care practice. However, GI and QHTMI scores were decreased in the smartphone application group. The reason behind this might be the constant notifications received in terms of oral health practice by the smartphone application, even though the participants were not in contact with their dentist. Therefore, participants from the test group were almost forced to take care of their oral health. However, both groups showed significant improvement in periodontal health after just one month. This result supported the outcomes of previous studies where periodontal health improved with time regardless of the methods used for oral hygiene instruction [21,34].

Gingival or periodontal health can be reliably measured by the GI [9,21] and QHTMI [24], and they have been used for many years in different studies. Hence, the current study also used similar GI and QHTMI scoring systems to assess periodontal health. Moreover, the current study ascertained significant differences in gingival health, which improved with time. Both GI and QHTMI score decreased in both the control and test group after the oral hygiene education, regardless of the method used for oral hygiene instruction. Moreover, using the advanced smartphone application showed a slight improvement in the GI and QHTMI score compared to those who received traditional instruction; however, that difference is not significant. This might be due to the eminence of the general knowledge the participants had regarding the importance of oral care. Details of oral hygiene instruction, including brushing time and techniques and use of dental floss and mouthwash, are generally taught to people in their childhood via different advertisements or in primary school. Due to the quality of the information the participants had already learned in different stages of their life, the use of advanced smartphone technology did not add any benefits to the patients. Socioeconomic status might have played an important role in the outcome of the current result; however, this variable was not assessed in the current study. A similar outcome was observed in a previous study where the WhatsApp messaging application was used to remind patients about their oral care practice [21]. In addition, a previous study on a brushing app (the Brush DJ app) showed that the app might be used as an important tool for the improvement of oral health; however, the app was more difficult to follow compared to the traditional lecture given on improving oral health [35]. The Brush DJ app is a commonly used app for dental hygiene and has been included in many previous studies [18,19,36,37,38]. This application plays a song of choice for two minutes during teeth brushing. The user can customize the background of the application in many ways and choose a song from a Spotify list [19]. Nevertheless, the current study did not use the Brush DJ app; however, some features of the application are similar. None of the patients complained about any difficulties in using the smartphone application that was used in this study. The application used in the current study was free and easily accessible to the current population which was one of the reasons for choosing the specific application. However, as per the literature search and best of our knowledge, none of the previous studies used the ‘Healthy Teeth–Tooth Brushing Reminder with timer’ application for oral hygiene instruction. Hence, no direct comparison could be performed of the effectiveness of using this application with the existing study.

On the contrary, some studies found promising results in using different applications in improving oral health care. A previous study with 400 students reported a significant decrease in QHTMI after six months in the test group compared to the control group [34]. In addition, patients with orthodontic treatment showed significantly lower bleeding index (BI), modified gingival index (MGI), and QHTMI in the text message group than the control group [39]. BI shows high specificity and sensitivity in periodontal health; therefore, the BI score is considered a convincing indicator of compliance with oral hygiene [40,41]. In addition, a decreasing score of GI indicates the improvement of gingivitis [42]. Therefore, assessing the GI and BI together in clinical trials should yield optimum results as these two components associate well [42]. However, the current study did not assess the BI; therefore, it is not prudent to directly compare its result with the aforementioned study. In addition, in the current study, no text messages were sent to the test group; only an automated notification from the application was received by the participants. Nevertheless, evaluating the BI along with the GI might alter the outcome of this study. Moreover, the periodontal condition is different in patients with or without orthodontic treatment. None of the patients in this study were undergoing orthodontic treatment. There was another study with autistic children that showed promising results in using the smartphone application [43], yet the current study did not include any patients with any systemic diseases. Therefore, the outcome could not compare directly with the current study.

Every study contains some limitations, and this study is not an exception. The sample size of the study was small and followed the convenience sampling technique due to the availability of participants. However, a larger sample size with proper sample size calculation would alter the outcome of this study. Concerning the sampling problem, this study included only male participants; therefore, a comparison of gender could not be performed. In addition, this study compared the periodontal condition only after one month. It would have been better if the comparison could have been performed in more than two stages. A longer duration might have altered the current outcome. Another factor is the age of the participants. The age range of the study was 18 to 60 years. However, the acceptance and understanding level of smartphone applications might vary with the younger generation compared to the older generation. Therefore, further clinical trial studies considering all the limitations would be recommended to draw a specific conclusion. Moreover, there are many smartphone applications available on the market with some strengths and weaknesses. It would also give a clearer insight into whether different applications could be compared in oral hygiene instruction in upcoming studies.

## 5. Conclusions

The current study did not find any evidence of the superiority of smartphone applications in periodontal oral health over the traditional method.

## Figures and Tables

**Table 1 ijerph-19-14703-t001:** GI and QHTMI scoring system.

GI Score	QHTMI Score
Score 0 = No inflammation	Score 0 = No plaque
Score 1 = Mild inflammation. Gingiva slightly changes in color and little change in texture	Score 1 = Separate fleck of plaque on the tooth
Score 2 = Moderate inflammation. Gingiva moderately glazing, redness, edema, and hypertrophy. Tendency to bleed upon probing	Score 2 = A thin continuous band of plaque
Score 3 = Severe inflammation. Marked redness and hypertrophy of gingiva. Tendency to spontaneous bleeding	Score 3 = A band of plaque up to one-third of the tooth
	Score 4 = Plaque covering up to two-thirds of the tooth
	Score 5 = Plaque covering two-thirds of the crown of the tooth.

GI: gingival index; QHTMI: Quigley–Hein Turesky modification index.

**Table 2 ijerph-19-14703-t002:** Comparison of periodontal status at T0 and T1 stages of the treatment between two groups.

Stages of Treatment	Periodontal Status	Groups	Mean	SD	*p*-Value
T0	GI	Control group	1.91	0.45	0.082
		Test group	2.08	0.69	
	QHTMI	Control group	2.77	0.48	0.059
		Test group	2.95	0.79	
T1	GI	Control group	1.54	0.37	0.125
		Test group	1.29	0.61	
	QHTMI	Control group	2.22	0.32	0.130
		Test group	2.01	0.53	

T0: before treatment; T1: after one month of treatment; GI: gingival index; QHTMI: Quigley–Hein Turesky modification index; SD: standard deviation.

**Table 3 ijerph-19-14703-t003:** Comparison of periodontal status between T0 and T1 stages of the treatment.

Group	Periodontal Status	Stages of Treatment	Mean	SD	*p*-Value
Control group	GI	T0	2.08	0.69	0.0001 *
		T1	1.28	0.61	
	QHTMI	T0	2.95	0.79	0.0001 *
		T1	2.01	0.53	
Test group	GI	T0	1.90	0.45	0.0001 *
		T1	1.54	0.37	
	QHTMI	T0	2.77	0.47	0.0001 *
		T1	2.23	0.32	

GI: gingival index; QHTMI: Quigley–Hein Turesky modification index; T0: before treatment; T1: after one month of treatment; SD: standard deviation; *; *p*-value < 0.05.

## Data Availability

The data used to support the findings of this study are available from the corresponding author upon request.

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
