# Peer review of "Effectiveness of Smart Applications of Oral Hygiene Instructions on Periodontal Health Compared to Oral Information: A Prospective Study"

_ijerph, 2022, doi:10.3390/ijerph192214703_

Round 1
Reviewer 1 Report
This manuscript requires intense effort to improve and develop. Therefore I have the following comments and suggestions, by answering and correcting, will support the strength of the manuscript. Please see the attachment.

Author Response
"Please see the attachment."

Reviewer 2 Report
Dear Authors,
I have been invited to review your work entitled “Effectiveness of smart applications of oral hygiene instructions 2 on periodontal health compared to traditional methods”. I believe it is a work of concern, however there are many major issues that deserve revision for the acceptance of this work to the IJERPH.
Please, provide a point-by-point response, highlighting the corrections with a color mark.
Title: you should add the study design in the title, that is “randomized clinical trial”. Conventional instructions ïƒ oral information
Abstract
- Add one sentence as an introduction.
- This current study ïƒ the present study
- Remove ICC statistics sentence.
- Please, change group A and B with Trial or Test and Control
Keywords
I suggest adding the following keywords: patients’ compliance
Introduction
- A short paragraph about social networks that have been also evaluated for the compliance of orthodontic patients could be added (https://doi.org/10.3390/app11020706).
Materials and methods
- If this is a randomized clinical trial, CONSORT guidelines should be followed. Therefore, reformat the manuscript according to the CONSORT paragraphs (take the cue from other studies on PubMed).
- Moreover, sample size calculation should be performed. I understand that there is no experimental drug used here, however sample size calculation helps selecting the right number of patients to obtain reliable statistically significant differences. You should add sample size calculation on PI or GI.
- Inclusion and exclusion criteria? For example, if the smartphone application can be used only on Android and iOS phones, phones adopting other operative systems could not install the application.
- Please, add the references for the GI and for the PI the first time you mention them. You should remove the scores of the two indexes if you add the reference.
Results
- Where is the intergroup comparison at T0? It should be performed to assess the homogeneity of the study sample.
Discussion
- Lines 149-154: a reference should be added for this paragraph.
Conclusions
- “Smartphone applications could encourage patients to maintain proper oral care.” This is not a conclusion of the present study. Please, focus only on the conclusions of the work.
Editorial issues
- English editing by a native speaker is recommended, spelling and editing errors should be corrected.
Author Response
"Please see the attachment."

Reviewer 3 Report
1. As the authors comment at the end of the discussion, one of the limitations of the study was the very small sample size, but how did you establish this sample size? Did you perform any kind of sample calculation? How did you consider 39 patients to be a sufficient size?
2. Table 1 is divided between two pages, it is best to keep it all on the same page, so you could reduce the font size of the table.
3. Why did the intra examiner calculation choose 8 patients randomly? Couldn't this have been done for all patients in the study, given the small sample size?
4. On page 4, line 182, the comma should be removed from the word floss. Also on page 5, line 188, the following should be revised: "similar outcome e observed...". What does the e stand for? Is it a drafting error?
5. The study time of 1 month is too short, a longer-term follow-up should be carried out to establish the reliability of the app's use.
6. Check the template of the journal to include the bibliographical references, as requested.
Author Response
"Please see the attachment."

Reviewer 4 Report
I would like to thank the editors of this journal for the opportunity to give my point of view on this interesting research project. On the other hand, I would like to congratulate the authors of this research, I consider that it is very necessary to investigate how to create good habits in dental hygiene in particular and in public health in general. I consider it very important to design initiatives in educational groups.
I believe that the study is well thought out and the methodology is appropriate, but I would like to make some comments that I believe could be addressed by the authors and improve some aspects of the study.
The theoretical framework does not consider group educational proposals in educational settings, where awareness of health issues has very good results.
https://doi.org/10.1111/josh.12388
https://doi.org/10.1097/00004650-199801000-00011
https://doi.org/10.1016/j.pec.2021.02.037
The study has worked with personal information from patients and has not indicated whether it has the report of the ethics committee.
I believe that the limitations section is brief and could be expanded, and would also benefit from a section on future uses of the study's conclusions.
I congratulate the researchers once again for the effort and dedication they have put into this great study and I thank the journal editors for the opportunity to have contributed my opinion on the research.
I consider the article to be publishable with minor changes.
Author Response
"Please see the attachment."

Round 2
Reviewer 1 Report
All comments were well answered except comment 4 and comment 5
Comment #4: (gingivitis or periodontitis) of the participants who were included in the study.
I would suggest giving more details about the periodontal condition in both control and experimental groups.
Reply to reviewer: Thank you so much for your comments. The corrections have been done as per comment.
Reviewer: I didn’t find the requested information; please add more details about the periodontal condition of both control and experimental groups, which means how was the periodontal condition of the participants before starting the examination.
Comment #5: In discussion, it was mentioned Though dentists provide the optimal treatment to the patients and enormous instruction on the chairside time on how to practice good oral care, it always depends on patients’ cooperation.. Since the author didn't mention the reference, I don't think this always depends on patients' cooperation.
I suggest deleting the word "always"
Reply to the reviewer: Thank you so much for your comments. The correction has been done as per the comment.
Reviewer: Why did you make all the responsibility for improving the oral hygiene to the patient? I suggested correcting the statement or at least deleting the word "always"
Author Response
"Please see the attachment."

Reviewer 2 Report
Dear Authors,
Thank you for providing the revised version of your manuscript.
There are some minor issues:
- I believe that Table 1 stands for T0 comparisons (and not T1, as you stated). You can merge table 1 and 2 into one table.
- I still don't understand table 3. Which comparisons are presented? What is the time frame (T)? In general, you should present a table with all the values per each T and then make the inter- and intragroup comparisons. Please, round to the two decimal places the data, except for p values.
- you do not want to remove the scores for PI and GI. I understand, however they are placed in a confusing way. Can you add a table, so that the manuscript is more clear?
Author Response
"Please see the attachment."

Reviewer 3 Report
Thank you for your reply.
Author Response
Dear Reviewer,
Thank you very much for your comments. The corrections have been done as per your comments.